# Scare-Away Risks: The Effects of a Serious Game on Adolescents' Awareness of Health and Security Risks in an Italian Sample

Francesca D'Errico *, Paolo Giovanni Cicirelli, Concetta Papapicco and Rosa Scardigno

Department of Educational Sciences, Psychology, Communication, University of Bari "A. Moro", 70121 Bari, Italy
* Correspondence: francesca.derrico@uniba.it

**Abstract:** Digital games can be defined as games supported by audiovisual apparatus and based on storytelling. This work aims to frame video games in the recent perspective offered by positive psychology and focuses on risk perception and on the promotion of protective behaviors in adolescent students, by means of a new Italian ad hoc created digital game named 'Scare-away risks' (Scacciarischi). In its storytelling, the different characters must defeat monsters that symbolically represent potential risks in home, school and work contexts. On this basis, the present study assumes that playing video games, as an engaging and motivating activity, can improve the adolescents' psychological perception of home, school and work risks. To verify these hypotheses, a quasi-experiment comparing students who played Scacciarischi (experimental group) with a control group was conducted. At the end of the game, a questionnaire was administered to 149 participants, balanced for gender, investigating (a) the level of engagement during the game, (b) the perceived risks, (c) the locus of control, (d) the protective behavior intentions. The results showed that playing Scacciarischi is related to higher levels of engagement, internal locus of control, risk perception and protective behavioral intentions. In addition, both engagement and internal locus of control act as precursors of the other two variables. Finally, reflections about the importance of the engaging experience of Scacciarischi in the domain of safety, prevention and health issues are proposed.

**Keywords:** serious game; engagement; risk perception; locus of control; protective behavior intentions

## 1. Introduction

Digital games, as a relatively new expression of the universal practice of playing games [1], can be defined as games "which we play thanks to an audiovisual apparatus and which can be based on a story" [2]. They represent a very popular leisure activity across the lifespan, usually based on the gamification techniques [3].

During the COVID-19 pandemic, the video game market boomed not only as a pastime, but as a possibility to connect with each other [4]. These data updated a core issue—widespread both in public opinion and scientific literature—concerning the balance between negative and positive effects of playing video games in human life, especially among young people.

Historically, a great amount of social research focused on the negative impact of gaming, claiming worries about the risk of increased aggression and desensitization to violence—especially for violent video games—and, more generally, about the potential harms related to addiction and depression. Actually, a recent study by Ferguson and Wang [5] suggested that early exposure to aggressive games was not a risk factor for later mental health symptoms—anxiety, depression, somatic symptoms, attention deficit hyperactivity disorder. In addition, in recent years a more "balanced" perspective emerged [6]. Especially when the target is represented by young players, since they have increasingly moved from solitary to multiplayer environments [7], scholars from developmental, social and media psychology focused on the gamers' psychological functioning [8], on the benefits of playing [6], on its potential to enhance mental health and wellbeing [9].

In the domain of gender studies, the classical literature about the "gender gap" reporting more "masculine" engagement [10] has been integrated by studies showing diminishing gender differences among young generations [11]. As a matter of this fact, gender has not been considered important in predicting, e.g., flow experience, even of gender can affect what aspects of a game can be considered enjoyable [12].

Nonetheless, even if no scientific evidence supported this claim, common beliefs are still anchored to gaming as a "boy thing", based on commonsense assumptions, such as boys being competitive, aggressive and undesirable individuals and girls lacking competitive attitudes and not enjoying violence in games [13]. Research on these matters also warns about the several consequences of these misogynistic targeting and stereotype threats, mainly identified in (a) different players' ability to create social connections through voice technology and feelings of distrust of un-known others [14]; (b) threatened female participation and performance in gaming, also related to the low self-identification as "legitimate" players [13]. As a consequence, females' experience reduced the possibility of pursuing professional careers, too.

In accordance with an interesting and inclusive review proposed by Granic et al. [6], the positive effects of playing video games can be organized in four main domains: cognitive, motivational, emotional, and social.

As for the cognitive benefits, a wide range of cognitive skills is promoted thanks to playing games, such as spatial skills [15], problem-solving [16], creativity [17] and so on. In recent years, self-efficacy also gained increasing attention [18], whereas less investigated were the internal locus of control and risk perception.

As for the motivational domain, facing multiple failures and celebrating the successfully completed challenging tasks have a great pulling-in strength. In addition, the possibility to obtain immediate and concrete feedback—such as coins, through points and so on—can both keep players within the "zone of proximal development" [19] (p. 86) and enable the acquisition of an incremental theory of intelligence—vs. an entity perspective.

The emotional benefits of playing are set in the theoretical background of the uses and gratifications theory, since playing preferred video games improves the players' mood as well as their positive emotions [20]. Among the deepest ones, players reported the well-known "flow" experience, that is to feel immersed in an intrinsically rewarding activity that improves a high sense of control and, at the same time, drops self-consciousness [21]. Psychological research about flow experience reported positive outcomes for adolescents: commitment and achievement in high schools, higher self-esteem and less anxiety were found [22].

As for the social domain, the immersive social contexts typical of video games can promote prosocial skills, especially when effective cooperation and helping behaviors are rewarded [23]. More importantly, these social skills might be generalized outside gaming, not only in peer and family relations but in the form of civic engagement [6].

In other words, running against the stereotype of gamers as "intellectually lazy, impulsive, and socially withdrawn", they must be "patient, focused, thoughtful, socially aware, and quick to respond physically and mentally" [4] (p. 1).

This work is set in the midst of discussions about the positive outcomes of playing video games, with a special focus on the so-called "serious games" as well as on their relations with specific psychosocial variables concerning cognitive, emotional and social domains. Whereas the scientific literature about these games is particularly concerned with the healthcare domain [24], we focused on the Italian serious game "Scare-away risks" (in Italian "Scacciarischi") especially devoted to the promotion of personal safety, risk perception and protective behaviors in ordinary contexts, such as home and schools. The overall aim of the project was to test if the Scare-away risks can improve the adolescents' awareness on safety issues by helping them to prevent risk behaviors, with early adolescence being a really critical developmental phase in the lifespan.

## 2. The Psychological Side of Gamified Educational Contexts

'Serious Games' or 'Game-Based Learning' are terms referring to digital play experiences whose purpose goes beyond mere entertainment and leisure; their broader goal is related to training, meaning and behavior change in different domains, such as education, healthcare, well-being, cultural heritage and so on [25]. With reference to traditional computer-based training, their peculiarity is to convey knowledge, skills and attitudes via play [2,25].

In the domain of healthcare, the use of video games tried to catalyze the motivational side, which is essential in patients, since they have to face either painful/aversive and boring/mundane practices. In addition, they propose an engaging and interactive distraction, involving multiple sensory systems which can explain the patients' involvement [24]. More generally, the efficacy of video games in this domain asks for their more widespread use; improving patients' participation, knowledge and adherence to treatment can promote higher quality care.

In the educational domain, since there is growing evidence that serious games can be effective learning tools, it is important to deepen which factors may impact successful learning [26]. Since "playing" is quite different from "working", students are more available to spend time with a game than in front of a (copy)book. Several reasons make video games "engaging", e.g., gaming can involve students who do not particularly "fit" with traditional learning strategies; in addition, rapid advances in information technology and the reduction of the attention span especially in the so called 'Net generation' are reshaping their learning styles [27].

Even if several definitions of engagement exist, it is mostly considered as a multidimensional construct, including behavioral, emotional and cognitive domains. In particular the so-called 'flow experience' can be considered the psychological starting point of the video-game players' engagement since this state refers to such exciting activities that anything else becomes oblivious for those who feel it [22]. Even if some negative features can be related to flow experience—such as reported anxiety, frustration, boredom and anger [28] and altered time perception [29], flow is mostly considered as a positive experience. In addition, even if scientific literature often related flow with enjoyment, flow is considered conceptually different from "immersion" (the first step in engagement), "presence" (a rather conscious state) as well as from "psychological absorption" and "dissociation" [30].

The affective experience of flow is related to the cognitive notion of self-efficacy [31], since users' emotional involvement and engagement are based on a positive perception about one's own capabilities. These beliefs about one's own ability to learn or perform specific activities at a certain level can concur to motivate adolescents to participate or not in those activities, implying either foregoing or persisting in the face of difficulties [28,32]. As applied to video gaming, self-efficacy concerns confidence in the ability to interact with video-game systems, and it is considered as one of the core factors making serious games successful [16].

Since playing video games is related to positive reinforcement, feedback and correction, on the one side self-motivation to continue is improved; on the other side, players are able to correct a mistake as well as to experience rewarded practices. Therefore, they can be aware of how they are doing the game and follow their progress. "This progress should reinforce their belief that they can overcome the next challenge. [ . . . ] Successfully completing each level of the game not only adds to the student's knowledge but also improves their perceived self-efficacy" [26] (p. 1455).

Whereas self-efficacy was quite investigated in social research about gaming, the impact of one of its core facets, that is locus of control, was less deep. Actually, internal locus of control, meant as the feeling that the experienced outcomes are mainly influenced by one's own actions, can be expected to be positively influenced by serious game playing [33]. Previous studies found associations between locus of control and learning performance in game-based learning, positive attitude toward learning and learning effectiveness [34].

Therefore, locus of control can work as an influential factor to predict a positive experience of learning.

Even if social research about flow and self-efficacy showed their general influence in learning [35–37], additional research seems necessary to better explain the relations between these constructs and serious games, especially in more neglected research areas.

For instance, as already testified in several healthcare domains, video gaming can take an essential function, that is to improve risk perception and preventive measures [37,38]. This domain represents a special challenge in actual times, since the COVID-19 pandemic unexpectedly impacted almost all communities around the world. In the theoretical background of constructivism [39], novices can learn from guidance and information, since knowledge is especially created through experience whilst exploring and performing activities. In order to create a kind of first-hand experience, "a safe but relatable virtual medium to construct the experience of how COVID-19 infection transmission occurs" [40] is needed.

In line with the Game Transfer Phenomena (GTP) [41], video game elements can be associated with real life thoughts, sensations and/or players' actions, both when occurring involuntary, automatically, and without premeditation by the players and when they intentionally integrate video games into their daily interactions. Therefore, this transfer can have a special relevance when risk perception and prevention are at stake.

Actually, psycho-social literature about risk perception was mostly oriented to investigate the relations between the degree of risk proposed in video games and the risk perception in real life, leading to controversial results: on the one side, generally no significant relations were found [42]; on the other side, some correlations existed in specific domains, such as reckless driving [43] and sport [43]. This last result can be explained as a consequence of the sense of invulnerability related to the perceived absence of negative consequences [44].

However, recently it was shown that: (a) media can considerably influence people's sense of risk—meant as the subjective assessment of the probability that possible negative consequences or diseases might come up—during the outbreak of public health emergencies [45]; (b) mass media exposure influenced the cognitive, emotional and preventive intention variables [46]. These interesting findings should be replicated in relation with the practice of playing video games.

More generally, despite the great enthusiasm about serious games, just a few were scientifically studied, meaning that it is not easy to specify the outcomes of gaming in comparison with other traditional educational approaches [6]. In addition, rather neglected was the study about the role of serious games in proposing virtuous practices and educating toward higher levels of risk perception and prevention, especially in early stages of life, as well as their efficacy in improving cognitive, emotional and social dimensions.

## 3. Materials and Methods

### 3.1. This Study

In line with the theoretical and empirical perspectives related to gamification, the general objective of this research was to verify whether the implementation of new technologies in the school context could represent an effective and engaging form of communication and learning. This overall aim both takes into account the main literature findings and tries to overcome some research gaps. As for the first side, since it is widely recognized that playing video games is related to positive effects in four main domains, that is cognitive, motivational, emotional, and social [6], this study analyzes specific variables pertaining to these dimensions. As for the second side, even if the literature about serious games is expanding, some features are still neglected:

(A)   The impact and effects of serious games have been investigated in several contexts, principally referred to education, healthcare, well-being, cultural heritage and so on [24–26]. To the best of our knowledge, there are no studies concerning dangers, risks and related practices in contextualized and everyday settings directly (home and school) or indirectly (work) experienced by adolescents;

(B)  As for the emotional and cognitive sides, a great attention has been devoted to engagement and self-efficacy and these factors have been considered somehow related [26]. However, the role of one of the core facets of self-efficacy, that is locus of control, has been less deepened, even if it can be considered a crucial dimension for attitudinal and behavioral outcomes [32,33];

(C)  As for the gender, even if recent findings tried to overcome the gender bias and stereotypes [11–14], additional studies are necessary to consolidate and to better comprehend eventual gender differences in experiencing and playing videogames.

Based on these shortcomings, we tried to investigate whether the psychological and educational sides that accompany the playful component of the serious game "Scare-away risks" (https://scacciarischi.it/it/ access on 1 July 2022) were able to reinforce cognitive aspects—awareness of risk factors at home, school and work and the associated internal locus of control—especially by means of emotional engagement experienced during the game, by improving also their intentions to assume protective behaviors.

More specifically, this study assumes that:

**Hp1**. *Playing Scacciarischi will improve the adolescents' engagement, support their awareness about the importance of one's own responsibility in several life domains, especially safety and health, as well as promote their risk perception and protective behavioral intentions associated with daily contexts.*

**Hp2**. *In line with the more updated literature, these variables will be empowered both in male and in female adolescents, thus overcoming traditional literature defining the activity of playing video-games as masculine-oriented.*

**Hp3**. *The level of engagement, together with the locus of control, can predict risk perception and protective behavioral intentions, implying that the higher the engagement and locus of control, the higher the risk perception and protective intentions will be found.*

### 3.2. Participants

The sample is composed of 149 adolescents, balanced for gender (50.3% males), aged from 11–13 (mean age 11.79) coming from several early secondary Apulian schools. The distribution across the two conditions of this study (which will be presented in the section "Procedure and measures") was almost equal (54.4% in Scare-away risks condition vs. 45.6% in control condition), as was the distribution across the different levels of the game (0 = 45.5%; 13.4% first level, 16.1% second level, 23.5% third level). Before taking part in the study, young participants' parents were asked to sign the informed consent. In addition, the study was conducted in accordance with the Declaration of Helsinki and approved by the Ethics Committee of University of Bari "A. Moro" (code: ET-22-01, 28 January 2022).

### 3.3. The Game

Scacciarischi is a videogaming brand realized by P.M. Studios, a gaming company, following the input promoted by the National Institute for Insurance of Accidents at Work-INAIL, agency of the Puglia region, and the Department of Health Promotion of Puglia region, with the collaboration of the Regional School Office of the same region. [https://scacciarischi.it/it/ access on 1 July 2022].

The aim of the project was to improve awareness among adolescents from several Apulian schools on safety and prevention issues in life and work contexts, specifically advantaging of the dynamic and engaging processes related to gamification. An additional extrinsic motivational level derives from the competition called "The Olympiad of Prevention", enabling participants to confront other players from the whole region as well as to win a prize that will be used to improve security in their schools.

Through Scacciarischi, participants can learn in a simple and direct way how to identify, prevent or manage risks and dangers at home (the first game level), at school (the second game level) and at work (the third game level). Each young player can choose among five avatars

representing the team of security heroes which, driving their "Safebots", fight against the evil "Dangerbots". These last ones have a symbolic configuration and concern the different risk domains, such as TeleMonster, MonsterSnack, EvilTube, Spruzzoveleni. The last introduced Dangerbot in the 2021 edition was devoted to the emergency of COVID-19, aiming to improve preventive behaviors during the pandemic.

Through a scenic design in 2D graphic (recalling the classical game Super Mario), Scacciarischi proposes three levels representing different life contexts (home, school and work), each one leading specific risks and dangers. The transition from one level to the next can happen thanks to several both mandatory and optional rewards (having again a symbolic configuration, such as facemasks and security tools), enabling the players to defeat the enemies as well as to gain lives/score (Figures 1 and 2). In addition, at the end of each level, the hero has to take down a harmful Dangerbot.

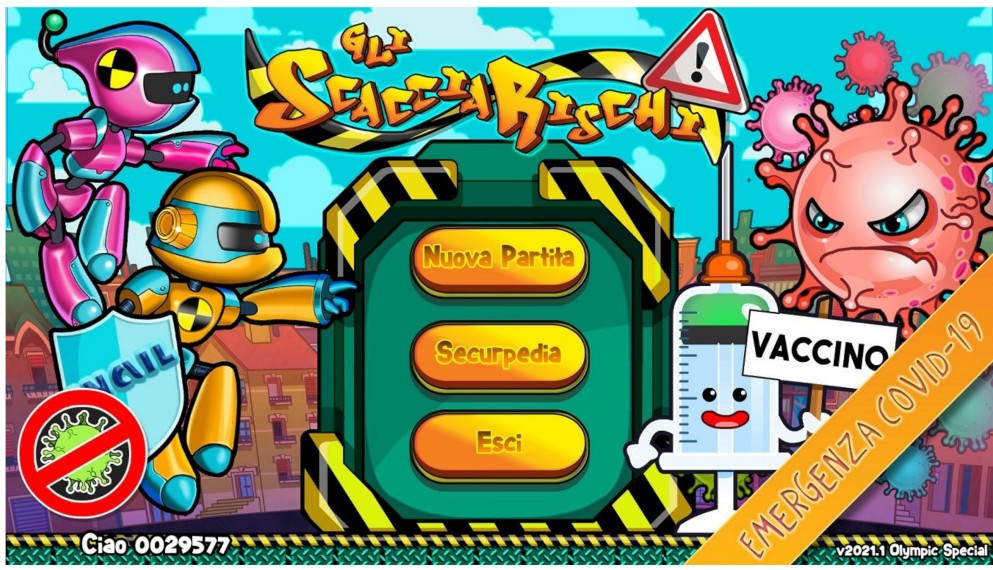

**Figure 1.** The starting interface of Scacciarischi.

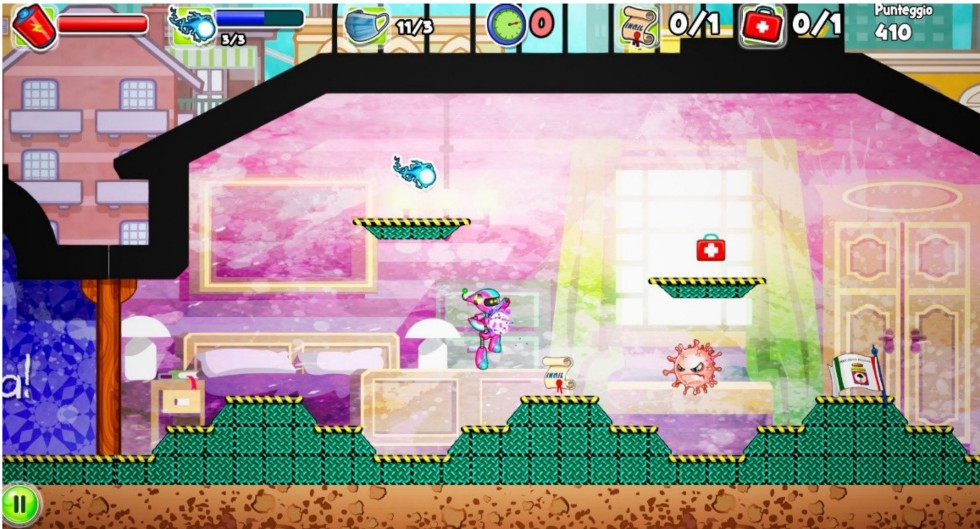

**Figure 2.** A scene from the first level of Scacciarischi.

A section of the serious game is dedicated to the "Securpedia", an encyclopedia of risks, which participants are required to consult before playing in order to overcome some game obstacles. In the passage from one level to the next, players have to answer some questions concerning risks, dangers and preventive behaviors. The Securpedia is structured in accordance with both the several contexts/game levels (home, school, work and COVID-19)

and the different school levels potentially adhering to the gaming sessions (primary and early secondary schools).

### 3.4. Procedures and Measures

In order to realize a quasi-experimental design, the following conditions were created: the experimental one, involving secondary school students who played with Scare-away risks for 3–4 months and answered the questionnaire referring to that game vs. the control one, composed by same-age students who answered the questionnaire by referring to their experiences with other video games.

As for the experimental condition, the project included two phases: (a) a training stage, in which participants could familiarize with the game; (b) the participation in inter-school Olympics, where they competed both individually and in teams. The players/students were supported by one or more teachers acting as tutors. At the end of the Olympics, the guide teachers were contacted to involve both playing Scacciarischi and not playing students of comparable age in our research. In this way, we obtained the experimental and control groups.

Within this quasi-experimental design, we controlled the effects of participation (or not) to the serious game Scare-away risks as well as of the gained game level on players' "engagement", "risk perception", "protective behavior", and "locus of control". We proposed an online questionnaire composed by the following measures, on a seven-point Likert-scale, adapted to our young sample (Appendix A):

- Engagement scale [47], with seven items having a sufficient reliability (Cronbach Alpha: 0.65). This scale conceptualized engagement as the contemporary presence of concentration, interest, and enjoyment. In particular, items concerned the following constructs: concentration, enjoyment, interest, challenge, skills, immersion and perceived learning. An example of an item measuring interest was: "How interesting was the game?"
- Risk perception scale, a five-item scale with a sufficient reliability (Cronbach Alpha: 0.64) conceptualizing risk perception as "people's negative estimation of possibilities of health issues or instances of which the disease can occur" [48] (p. 141). The reworked items of this scale referred to the risks as encountered in the game (e.g., "I know that toxic substances are present at home");
- Protective behavioral intentions scale, composed of five items showing a high reliability (Cronbach Alpha: 0.83). This variable was defined through a set of behaviors useful to protect against risks, referring to the different game contexts/levels, for example "Wash your hands regularly with water and soap for 40 s";
- Locus of control scale [49], with five items showing a sufficient reliability (Cronbach Alpha: 0.69). This construct was referred to the individual's beliefs that events in life (in this case concerning health) are produced by his/her own behaviors or actions (or, on the opposite pole, by external causes, independent from his/her will). So, items such as "My physical health depends on my closest friends" or "My physical health depends largely on what I do (and I don't)" were included.

Participants of the control group were asked to answer a questionnaire composed of the same number of items (and same Likert points), referring to other video game experiences, with no references to Scacciarischi and its levels.

Table 1 summarizes all the variables included in this study.

**Table 1.** Dependent (DV), independent (IV), and co-varying (CV) variables of the quasi-experiment.

| Dependent Variables | Independent Variables | Co-Varying Variables |
|---|---|---|
| Engagement | Experimental Condition | |
| Risk Perception | Gender | |
| Protective Behavioral | Game level | Locus of Control |
| Intention | Gender | |

## 4. Results

A multivariate ANOVA between subjects (experimental/control conditions and gender) pointed out significant differences in engagement [$F_{(1, 146)}$ = 5.19; $p < 0.024$] and risk perception [$F_{(1, 146)}$ = 14.12; $p < 0.00$], meaning that the mean scores were higher in the Scacciarischi condition compared to the control one for all the considered dimensions (see Table 2). Referring to gender, the ANOVA analysis revealed a main effect for all the dependent variables. Rather unexpectedly, girls were more engaged [$F_{(1, 146)}$ = 2.76; $p < 0.05$].

**Table 2.** (**a**) Mean and standard errors for experimental and control conditions. (**b**) Mean scores and standard errors for gender.

| (a) | | | |
|---|---|---|---|
| **Dependent Variables** | **Condition** | **Mean** | **Standard Error** |
| Engagement * | Scacciarischi | 3.03 | 0.062 |
| | Control | 2.88 | 0.068 |
| Risk perception * | Scacciarischi | 4.49 | 0.082 |
| | Control | 4.23 | 0.090 |
| Protective behavioral intentions | Scacciarischi | 4.11 | 0.083 |
| | Control | 3.91 | 0.091 |
| (b) | | | |
| **Dependent Variables** | **Gender** | **Mean** | **Standard Error** |
| Engagement * | Boy | 2.84 | 0.064 |
| | Girl | 3.08 | 0.065 |
| Risk perception * | Boy | 4.18 | 0.087 |
| | Girl | 4.53 | 0.090 |
| Protective behavioral intentions | Boy | 3.95 | 0.087 |
| | Girl | 4.07 | 0.088 |

* statistically significant.

In addition, the locus of control being a personality trait variable, we checked it as a covariate in the analysis and found it is significant across all the three independent variables (engagement [$F_{(1, 146)}$ = 13.64; $p < 0.000$, risk perception [$F_{(1, 146)}$ = 68.84; $p < 0.000$ and protective factor [$F_{(1, 146)}$ = 75.21; $p < 0.000$]). This means that the higher participants' locus of control, the higher their level of engagement is during the game and the higher their risk perception.

As for the game level, running a multivariate ANOVA with the level of the game and gender, with locus of control as a covariate, significant differences were found in the following direction: in relation with higher game levels, higher scores for engagement in game [$F_{(1, 146)}$ = 4.16; $p < 0.05$], risk perception [$F_{(1, 146)}$ = 3.09; $p < 0.05$], internal locus [$F_{(1, 146)}$ = 2.32; $p < 0.04$] and protective behavioral intentions [$F_{(1, 146)}$ = 4.09; $p < 0.03$] were found (Table 3; Figure 3).

**Table 3.** Game Levels and variables mean scores.

| Game Level | Engagement | St. Error | Risk Perception | St. Error | Protective Intentions | St. Error |
|---|---|---|---|---|---|---|
| 0 | 2.88 | 0.06 | 4.23 | 0.09 | 4.11 | 0.09 |
| 1 | 2.85 | 0.11 | 4.28 | 0.16 | 3.79 | 0.16 |
| 2 | 2.99 | 0.11 | 4.45 | 0.16 | 3.82 | 0.16 |
| 3 | 3.25 | 0.09 | 4.69 | 0.12 | 4.09 | 0.12 |

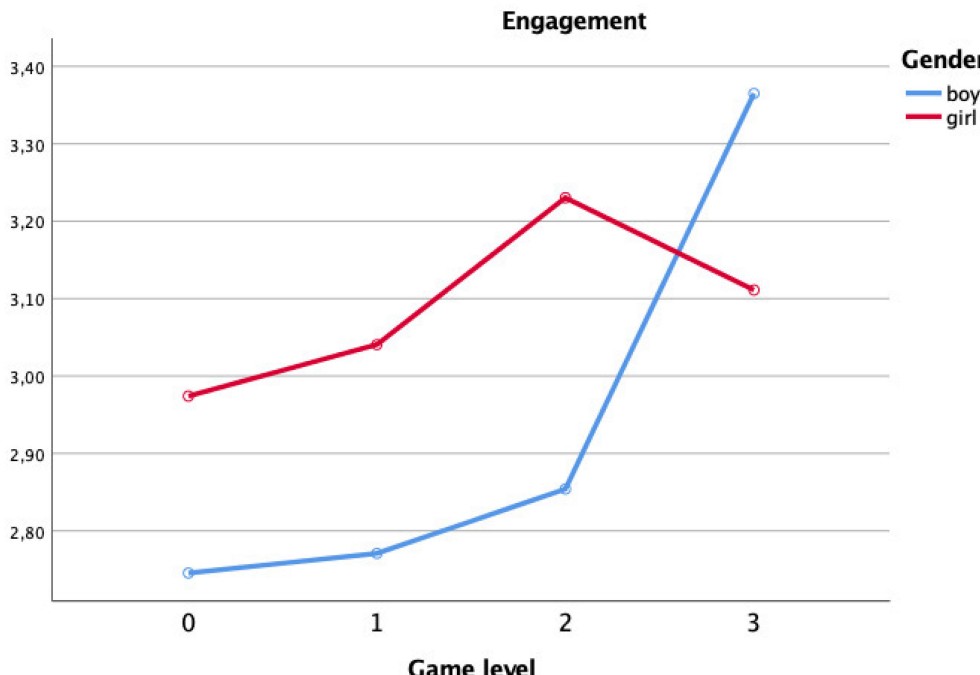

Le covariate presenti nel modello sono valutate ai seguenti valori: internal_locus = 3,8935

**Figure 3.** Experimental Condition*Gender on Engagement.

Additionally, gender significantly contributes to both level of engagement [$F(1, 146) = 2.46$; $p < 0.05$] and risk perception [$F(1, 146) = 3.80$; $p < 0.025$] (Figures 3 and 4). Specifically, the post-hoc Tukey test pointed out, for engagement and risk perception, respectively, that the strongest difference resulted for engagement between level 0 and first with the third level ($p < 0.05$), as well as for risk perception when comparing the group 0 with group who played at the third level ($p < 0.05$). This means that the higher level of the game played, the higher the level of engagement and risk perception reported by participants.

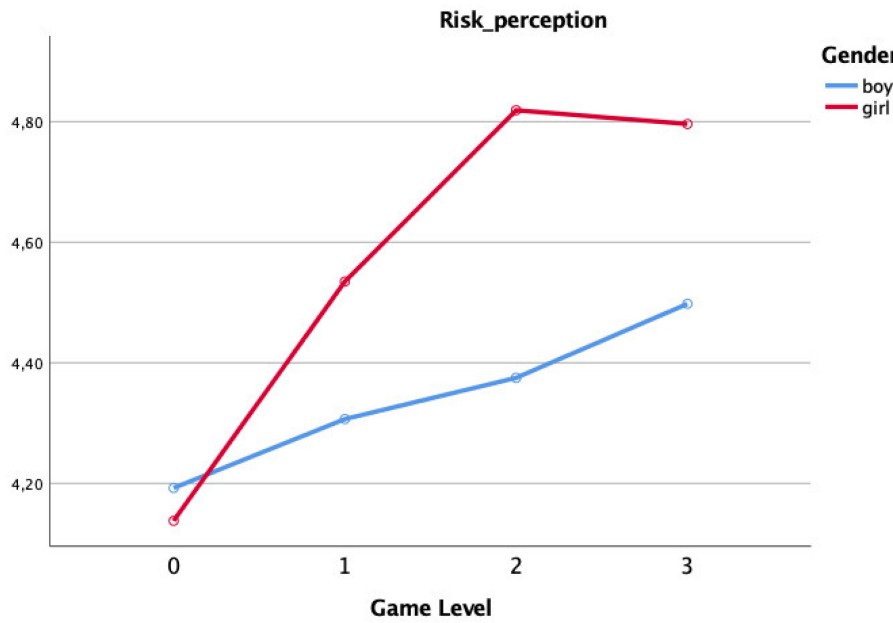

Le covariate presenti nel modello sono valutate ai seguenti valori: internal_locus = 3,8935

**Figure 4.** Experimental Condition*Gender on Risk Perception.

Furthermore, the ANOVA pointed out a significant interaction effect between gender and level of the game [F(1, 146) = 3.09; *p* < 0.05]: whereas girls experienced higher levels of engagement in the early game levels, boys obtained higher ones only at the last level (Figure 3).

To resume, the positive emotional state experienced during the game is a good condition to reinforce the other dimension related to the risk, protective behavioral intentions and internal locus, as demonstrated by Pearson correlations (Table 4).

**Table 4.** Correlations among engagement and risk perception, internal locus of control and protective behavioral intentions.

| | | Internal Locus | Risk Perception | Protective Intentions |
|---|---|---|---|---|
| engagement | Pearson correlation | 0.308 ** | 0.441 ** | 0.382 ** |
| | Sign. (two-tailed) | 0.000 | 0.000 | 0.000 |

** statistically significant.

In particular, together with the sense of control represented by the adolescents' internal locus, the level of engagement impacted significantly on the reinforcement of young players' risk awareness (Table 5) and on protective behavioral intentions (Table 6), as reported by the indexes obtained by the linear regression below.

**Table 5.** Linear Regression of Engagement and Locus of Control Predicting Risk Perception. Results of linear regression analysis examining the effect of engagement and internal locus in predicting children's risk perception.; b: unstandardized regression coefficient; SE: standard error; β: standardized coefficient; t: obtained t-value; p: probability; R2: proportion of variance explained.

| | b | SE | β | t | p | R2 |
|---|---|---|---|---|---|---|
| Constant | 1.346 | 0.32 | | 35,150 | 0.00 | |
| Engagement | 0.409 | 0.093 | 0.29 | 4019 | 0.00 | |
| Internal locus | 0.465 | 0.070 | 0.19 | 2717 | 0.007 | |
| Model | | | | | | 0.00 |

**Table 6.** Linear regression of Engagement and Locus of Control Predicting Protective behavioral intentions. Results of linear regression analysis examining the effect of engagement and internal locus in predicting children's protective behavioral intentions; b: unstandardized regression coefficient; SE: standard error; β: standardized coefficient; t: obtained t-value; p: probability; R2: proportion of variance explained.

| | b | SE | β | t | p | R2 |
|---|---|---|---|---|---|---|
| Constant | 1.105 | 0.314 | | 48,190 | 0.00 | |
| Engagement | 0.291 | 0.089 | 0.219 | 3244 | 0.001 | |
| Internal locus | 0.525 | 0.67 | 0.528 | 7817 | 0.000 | |
| Model | | | | | | 0.00 |

As for the risk perception as a dependent variable, linear regression revealed that both engagement and internal locus of control acted as precursors of risk perception for young players (Beta:0.19)—this model being significant (t(2, 146) = 4.01; *p* < 0.00). The results from regression (Table 6) also emphasized a very significant influence of internal locus of control on higher protective behavioral intentions—as tested by the beta level reported below.

## 5. Discussion

This work tried to overcome some limitations deriving from the attractive and promising "gamified" technologies: even if the increasing impact of serious games is widely acknowledged, especially in healthcare and educational domains, a systematic analysis of the possible benefits of gaming is rather at the beginning, especially considering the literature on young gamers' risk awareness and their safety behaviors in ordinary contexts and experiences, such as at home, school and work. In addition, we proposed both well-explored variables (e.g., engagement) and some neglected ones, such as locus of control, in order to investigate their relations with attitudinal and behavioral dimensions.

Our general aim was to investigate some specific psycho-social effects of the implementation of new technologies in the educational domain. In particular, the serious video game Scacciarischi (in English "Scare-away risks") was proposed in several early secondary schools in the South of Italy to raise awareness among children and adolescents about safety and prevention issues. In this study, this experimental condition was compared with students of a control condition who did not play Scacciarischi but could report other video game experiences. The questionnaire proposed to participants was composed of four main measures, that is, "engagement", "locus of control", "risk perception" and "protective behavioral intentions", which were adapted with reference to both young participants and the video game features.

A first, general and interesting results were related to the analysis concerning the two independent variables, experimental condition and gender, controlling for the perceived locus of control. The level of engagement and risk perception were significatively higher for young gamers and especially for girls. This first result confirms the efficacy of serious video games in promoting improvements in the cognitive and emotional domains, as emphasized by the literature [6].

In particular, from the emotional side, those who played Scacciarischi felt more concentrated, careful, amused and involved if compared with those who did not (but were playing other games); from the cognitive side, the experimental condition enabled participants to be more focused on their subjective power to produce and/or control some events of their life especially dealing with health matters; even risk perception was improved in players, showing higher levels of awareness and care in all the domains expressed by video games (home, school and workplace). As for the protective behavioral intentions, higher scores were found, and this means that young participants seem to be ready to employ protective measures across several contexts (home, school and so on).

The holistic and general benefits deriving from playing Scare-away risks were confirmed by two additional results. Firstly, the highest level obtained in gaming was accompanied by the highest scores for both engagement and risk perception. In particular, both variables increased level by level, even if the most significant enhancement was found between the first and the third level, as showed in the Tukey post-hoc. As for the other variables, the highest level obtained was higher than the previous ones, but the score in the final level was similar to the control condition. This means that future studies will better test these two variables by improving our scales with more specifications to the game actions.

Second, significant and high correlations were found among the variables, and in particular the experienced engagement was significantly correlated with the other three. These results can be explained as follows: (a) since engagement can be considered as a motivational and emotional dimension, it is related to uses and gratifications theory [20] as well as to several positive outcomes [21,22]; therefore, it increases when video games fit their motivational and emotional needs, offering progression goals and feedback; (b) the increasing scores in risk perception can be especially related to the game content, having to do with risk factors across several contexts; (c) all the involved domains—cognitive, emotional and motivational—were related to each other. These results can be considered as an extension of the Game Transfer Phenomena [41] in the domain of safety and prevention, meaning that video games are integrated into their daily perceptions and interactions [42].

A confirmation about this last concern was offered by the linear regressions: both engagement and internal locus of control significantly predicted the cognitive variable 'risk perception' and also the behavioral intentions of protecting one's own safety. In other words, these variables act as emotional, motivational and cognitive "scaffolds" [50] and support traditional cultural learning tools, cognitive perceptions and behavioral practices. As an engaging and educative tool, the video game Scacciarischi appeared to be able to improve the perception change and behavior monitoring in several domains, expanding the results obtained by other ones related, e.g., to COVID-19 [41]. These results confirmed the Technology Acceptance Model (TAM), which included the variables of enjoyment and perceived self-efficacy as predictors of system use and learning outcomes [41,51,52]. In addition, as these models mostly focus on engagement and self-efficacy as predictors, our study introduced the (internal) locus of control, meant as the feeling that the experienced outcomes are mainly influenced by one's own actions, specifically oriented in the health domain [50], as a factor predicting risk perception and protective behavior.

Additional interesting results derive from the "gender" variable. Despite the unstable oscillation between classical literature (finding boys as more motivated to play games, to enjoy the excitement and fantasy aspects), more recent studies (showing diminishing gender differences among young generations) [11] and common beliefs still anchored to gaming as a "boy thing" and menacing the equal participation in game activities [13,14], this study offers new perspectives about gender studies, since female participants experience more "engagement" in gaming if compared with males. In line with other findings, if video games were traditionally considered as a "realm dominated by males" [10] (p. 911), this gender gap is considered overcome by a gender-specific adaptations approach. Looking specifically at the gender engagement across the levels, the general result is confirmed in the three lower levels, while in the last level the situation is switched, and therefore boys showed higher involvement. On the one side, this result can support the efficacy of Scacciarischi in contrasting the "gender gap" mostly founded in archaic gender role portrayals, presence of violence in games, lack of social interactions and competitive elements; on the other side, the crossing in the higher level can somehow support the different gender media literacy and affinity for information technologies.

The strengths of this work are related to the opportunity to study psychosocial effects of the application of the intriguing language of serious games to the educational domain of health and safety. Our work made use of both well-known variables in the literature about video gaming (such as engagement) and more neglected ones (e.g., internal locus of control, concerning health issues). Scacciarischi was created to involve young boys and girls in new learning dynamics concerning health issues, safety and risks across several contexts. Therefore, this program had a special responsibility in the COVID-19 pandemic, since health and safety were in the foreground and in line with the special needs to educate younger generations.

Nonetheless, several limitations can be presented: first, as cross-sectional research, no causality relations can be claimed; second, we gathered data after the playing sessions, therefore it is not possible to explore the longitudinal effect of the game; third, some results need to be deepened also by taking into account the level of participants' addiction to video games. In order to better comprehend the implications of our findings for boys' and girls' education, training and developmental opportunities, further investigation should be proposed, e.g., involving more "social" variables.

**Author Contributions:** Conceptualization, F.D. and R.S.; methodology, F.D., R.S., P.G.C. and C.P.; data analysis, F.D. and P.G.C.; writing—original draft preparation, R.S., C.P. and F.D.; writing—review and editing, R.S.; supervision, F.D. All authors have read and agreed to the published version of the manuscript.

**Funding:** This research received no external funding.

**Institutional Review Board Statement:** The study was conducted in accordance with the Declaration of Helsinki, and approved by the Ethics Committee of University of Bari "A. Moro" (code: ET-22-01, 28 January 2022).

**Informed Consent Statement:** Informed consent was obtained from all the parents' subjects involved in the study.

**Data Availability Statement:** The data that support the findings of this study are available from the corresponding author upon reasonable request.

**Acknowledgments:** Special thanks to all the partners of the Scacciarischi project, financed by Inail Puglia, as part of a motivated and multifaceted work team, in particular Lorenzo Cipriani (Inail Puglia), Fabio Belsanti (CEO & Lead Game Designer at Age of Games), and Region of Puglia (It).

**Conflicts of Interest:** The authors declare no conflict of interest.

## Appendix A

Answer the following questions on a scale from 1 to 5, where 1 indicates that you strongly disagree and 5 that you strongly agree

Socio-demographics

(1)    How useful is it to use video games to acquire new knowledge?

Locus of control

1.    My physical health depends on my closest friends
2.    I feel that my physical health is something that depends on myself
3.    My physical health largely depends on what I do (and don't do)
4.    If I am physically well, I owe it to my parents (*)
5.    Staying in good physical health is the result of my commitment and my skills

Engagement

(1)    The videogame helped me to learn new contents.
(2)    How difficult was it to concentrate? (*)
(3)    The game delivered contents that caught my attention.
(4)    How much fun did you have during the gaming activity?
(5)    I was so involved in the game that I forgot other things.
(6)    How difficult was the game for you?
(7)    I felt prepared during the game. (*)

Perception of risk

(1)    Covid is easily transmitted when we are in close range of other people
(2)    I know there are toxic substances in the house
(3)    I think that an incorrect body position, even while sitting at the desk, can cause physical problems
(4)    In the construction of a site, electric current represents a danger
(5)    Uncontrolled information can hide lies

Protective measures and behaviors

(1)    Wash your hands regularly with soap and water for 40 s
(2)    It is important to put the games back in their place, especially if they are close to steps and stairs
(3)    I should notify the teacher of any incident, even if it seems unimportant
(4)    At work, the (protective) helmet protects against falling materials and tools from above
(5)    If I think a piece of news is false, I can ask for help from adults I trust

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
