# Peer review of "Scare-Away Risks: The Effects of a Serious Game on Adolescents’ Awareness of Health and Security Risks in an Italian Sample"

_mti, doi:10.3390/mti6100093_

Round 1

Reviewer 1 Report

The article 'Scare-away risks: the effects of a serious game on adolescents' awareness of health and security risks' deals with a topic of great interest to the scientific community. The topic is central because it has been approached from a different perspective to capture the positive aspects of video game use.

The article is well written and the methodological choices pertinent. The need to consider alpha indices below .70 as 'sufficient' rather than good is noted. 

Author Response

Thank you for your suggestion, we modified in the text. The authors

Reviewer 2 Report

TITLE: given that the student sample is all made of Italian students, this should be reflected in the title.

ABSTRACT

An abstract should not contain any references. It is suggested to make a sentence describing digital games rather than reporting the reference.  

LITTERATURE REVIEW

Some of the literature utilised in the review is over 15 years old and should really be replaced, as some of the findings have moved on. Only iconic paper older than 10 years (2012) should be kept.

For example, the quote paper on game and gender differences is 11 years old:

  1. Hartmann, T. and Klimmt, C. Gender and computer games: Exploring women’s dislikes. Journal of Computer-Mediated Commu- 480 nication 201111, 910-931. doi: 10.1111/j.1083-6101.2006.00301.x

  2.  

 and research has moved on, for example:

Sánchez-Mena, A., Martí-Parreño, J., & Aldás-Manzano, J. (2019). Teachers’ intention to use educational video games: The moderating role of gender and age. Innovations in Education and Teaching International56(3), 318-329.

Madden, D., Liu, Y., Yu, H., Sonbudak, M. F., Troiano, G. M., & Harteveld, C. (2021, May). “Why Are You Playing Games? You Are a Girl!”: Exploring Gender Biases in Esports. In Proceedings of the 2021 CHI Conference on Human Factors in Computing Systems (pp. 1-15).

Vella, K., Klarkowski, M., Turkay, S., & Johnson, D. (2020). Making friends in online games: gender differences and designing for greater social connectedness. Behaviour & Information Technology39(8), 917-934.

MATERIAL & METHODS

This section is more of mixed information that should instead be organised into sub-sections.

First, the paper should have a section named "This Study" or a similar name, in which the researchers summarise the literature findings and highlight the research gap (one short paragraph). Following they should express the exact research question and hypothesis. There are some elements of the above in the section Material & methods (line 182), which should be placed in a section on their own. In particular, it is suggested to add a table highlighting the Dependent (DV), independent (IV), and co-varying (CV) variables of the experiment.

Hp1: Please note that Locus of control is a personality trait, and claiming that playing the game only for a short period of time might change one's personality is a bug claim, which the reviewer suggests revising.

Hp2: this should be revised in light of the more recent literature, for example in the UK, female gamers are in number similar to males.

The participant's description should be a subsection of the method section.

The Game Schiacciarischi should also be described in a sub-section of the Material & Methods section, including why the game was created. A walkthrough of the age should be provided to aid the reader in understanding what the game is about. Also, there should be a detailed explanation of the game sued in the control group.

The experimental procedure should be described in a sub-section own its own of the Method section.

Similarly, each tool utilised (questionnaires) should be in a sub-section of the method section.

The Ethical approval for the study should be reported.

ANALYSIS

Each hypothesis should have its own subsection under the Analysis section.

In particular, Hp1 has been analysed with a Multivariate, and the table with the various variables (IV, CV, DV)  requested above should help clarify how the multivariate has been run, which at the moment is obscure(but see reviewers' suggestion below). Also, the post-hoc test should be reported as a table. In particular, Gender is a variable that has been manipulated (data is gender balanced), thus gender should be listed included in the Hp1, and listed as IV.

Perhaps the multivariate test should consider the following:

IV: experimental group vs control, gender

CV: Locus of control (this is a personality trait that can change the effect on the DV)

DV: engagement, protective behavioural intentions, risk perception.

Hp3: This question was run as a regression, which is correct, but it is not clear if this is a multi-variable or a single variable linear regression. From the table 4 it appears this is a multiple linear regression (hierarchical or all together?). However, please note that linear regression is perhaps redundant after running the multivariate suggested above. It is playing the game type (experimental or control)  that changes risk perception, as game engagement depends on the type of game played. 

Also, Considering Locus of control as CV, allows to see its effect on risk perception.

finally, a table reporting a Pearson correlation of all the variables considered should be reported, as it is useful to the readers, but perhaps still redundant given the multivariate east above

DISCUSSION

This should be revised in light of the new Multivariate to be run as suggested above and set in the context of the existing literature highlighting the novel findings.

Author Response

Dear Reviewers,

Firstly, we would like to thank you for your attentive and ameliorative suggestions. We took into account any observation and, in this way, our work has been widely integrated and improved.

Below you will find our answers for each point.  

TITLE: given that the student sample is all made of Italian students, this should be reflected in the title.

A: The title has been so modified: “Scare-away risks: the effects of a serious game on the adolescents' awareness of health and security risks in an Italian sample”

ABSTRACT

An abstract should not contain any references. It is suggested to make a sentence describing digital games rather than reporting the reference.  

A: We removed the reference in the abstract. As a consequence, the current first two references have been inverted.

LITTERATURE REVIEW

Some of the literature utilised in the review is over 15 years old and should really be replaced, as some of the findings have moved on. Only iconic paper older than 10 years (2012) should be kept.

For example, the quote paper on game and gender differences is 11 years old:

  1. Hartmann, T. and Klimmt, C. Gender and computer games: Exploring women's dislikes. Journal of Computer-Mediated Commu- 480 nication 201111, 910-931. doi: 10.1111/j.1083-6101.2006.00301.x

 and research has moved on, for example:

Sánchez-Mena, A., Martí-Parreño, J., & Aldás-Manzano, J. (2019). Teachers' intention to use educational video games: The moderating role of gender and age. Innovations in Education and Teaching International56(3), 318-329.

Madden, D., Liu, Y., Yu, H., Sonbudak, M. F., Troiano, G. M., & Harteveld, C. (2021, May). "Why Are You Playing Games? You Are a Girl!": Exploring Gender Biases in Esports. In Proceedings of the 2021 CHI Conference on Human Factors in Computing Systems (pp. 1-15).

Vella, K., Klarkowski, M., Turkay, S., & Johnson, D. (2020). Making friends in online games: gender differences and designing for greater social connectedness. Behaviour & Information Technology39(8), 917-934.

A: Most of the research articles older than 10 years were replaced. In some cases, we found updated works by the same authors (e.g. Nakamura, J., & Csikszentmihalyi, M. 2014); in other ones new works proposing definitions, concepts or results were found (e.g. the uses and gratification background). We just left articles older than 10 years in case of general definitions and/or well recognized papers.

As for the gender differences, we really thank you for your useful suggestions. We made use of them and additional ones. The reference to Hartmann was kept just to say it has been overcome.   

MATERIAL & METHODS

This section is more of mixed information that should instead be organised into sub-sections. First, the paper should have a section named "This Study" or a similar name, in which the researchers summarise the literature findings and highlight the research gap (one short paragraph). Following, they should express the exact research question and hypothesis. There are some elements of the above in the section Material & methods (line 182), which should be placed in a section on their own. In particular, it is suggested to add a table highlighting the Dependent (DV), independent (IV), and co-varying (CV) variables of the experiment.

Hp1: Please note that Locus of control is a personality trait, and claiming that playing the game only for a short period of time might change one's personality is a bug claim, which the reviewer suggests revising.

Hp2: this should be revised in light of the more recent literature, for example in the UK, female gamers are in number similar to males.

A: Hp2 has been revised in light of more update literature.

The participant's description should be a subsection of the method section.

The Game Schiacciarischi should also be described in a sub-section of the Material & Methods section, including why the game was created. A walkthrough of the age should be provided to aid the reader in understanding what the game is about. Also, there should be a detailed explanation of the game sued in the control group.

The experimental procedure should be described in a sub-section own its own of the Method section.

Similarly, each tool utilised (questionnaires) should be in a sub-section of the method section.

A: We followed all these suggestions and we better organized the section “material and methods” in the following sub-sections:

  1. a) this study (presenting the overall aim, the literature findings/shortcomings and our hypothesis);
  2. b) participants (where the sample is better presented)
  3. c) the game (Scacciarischi has been more widely described and contextualized)
  4. d) procedure and measures (where the experimental procedure, the control group and the measures have been specified). In this subsection the table including all the variables was added too.

The Ethical approval for the study should be reported.

A: We reported it also in the text of the manuscript (we had already reported it in the final declarations).

ANALYSIS

Each hypothesis should have its own subsection under the Analysis section.

In particular, Hp1 has been analysed with a Multivariate, and the table with the various variables (IV, CV, DV) requested above should help clarify how the multivariate has been run, which at the moment is obscure(but see reviewers' suggestion below). Also, the post-hoc test should be reported as a table. In particular, Gender is a variable that has been manipulated (data is gender balanced), thus gender should be listed included in the Hp1, and listed as IV.

Perhaps the multivariate test should consider the following:

IV: experimental group vs control, gender

CV: Locus of control (this is a personality trait that can change the effect on the DV)

DV: engagement, protective behavioural intentions, risk perception.

A: We reported in the paper the suggested multivariate, including gender among IV and locus of control as covariate. It allows us to improve the significance of variable as in the case of risk perception thus we also include this new significant result and when possible the post-hoc. In the paper at lines 355-435 we corrected tables, figures and the description of the results.

Hp3: This question was run as a regression, which is correct, but it is not clear if this is a multi-variable or a single variable linear regression. From the table 4 it appears this is a multiple linear regression (hierarchical or all together?). However, please note that linear regression is perhaps redundant after running the multivariate suggested above. It is playing the game type (experimental or control) that changes risk perception, as game engagement depends on the type of game played. 

Also, Considering Locus of control as CV, allows to see its effect on risk perception.

finally, a table reporting a Pearson correlation of all the variables considered should be reported, as it is useful to the readers, but perhaps still redundant given the multivariate east above

A: Thank you for your question that helps us to better explain the analysis; It is a linear regression with all the variables taken together. We fairly agree with your point concerning the redundancy of the table 4 but we prefer to maintain the table 4 since it is focused only on the participants who played the game by reinforcing the results discussed below where we essentially aimed at explaining the difference across experimental conditions.  

DISCUSSION

This should be revised in light of the new Multivariate to be run as suggested above and set in the context of the existing literature highlighting the novel findings.

A: Done. Thank you.

Reviewer 3 Report

First of all, I would like to congratulate the authors of this manuscript.

In spite of the fact that this manuscript is written well, I have some questions:

Do you have information about whether the participants are "game-addict" or not? I mean do you know how many hours they are playing usually video games per week? Please write some sentences about it in the "3. Materials and Methods" section.

Can you show the online questionary maybe in an appendix section?

To sum it up, after answering the above-mentioned questions, I suggest this manuscript for publication.

Author Response

Thank you for your comments. Unfortunately, we did not measure how much participants were addicted but mainly how much they evaluated useful playing video-game. We added in the discussion this as a limit to take into account for future studies (row 539). We also added an appendix to the paper. Best, the authors 

Reviewer 4 Report

The article presents an in-depth study conducted on adolescents regarding the game Scacciarischi and outlines the problems encountered during the game.

The study is very well organised and conducted. Two sets of children were used: one played the game, the other did not.

Gli autori hanno affrontato il problema dell'apparente diversa attitudine tra sessi diversi a svolgere queste attività e partecipare ai giochi.

L'articolo è ben scritto e strutturato

Author Response

Dear reviewer, thank you for your reading and words of appreciation. 

the authors